# Linear Elastic Fracture Mechanics Assessment of a Gas Turbine Vane

**DOI:** 10.3390/ma15134694

**Published:** 2022-07-04

**Authors:** Blanca Orenes Moreno, Andrea Bessone, Simone Solazzi, Federico Vanti, Francesco Bagnera, Andrea Riva, Daniele Botto

**Affiliations:** 1Department of Electronics and Telecommunications, Politecnico di Torino, 10129 Torino, Italy; orenesmb@gmail.com; 2Turbine Mechanical Integrity Department, Ansaldo Energia, 16152 Genova, Italy; andrea.bessone@ansaldoenergia.com (A.B.); simone.solazzi@ansaldoenergia.com (S.S.); federico.vanti@ansaldoenergia.com (F.V.); francesco.bagnera@ansaldoenergia.com (F.B.); andrea.riva@ansaldoenergia.com (A.R.); 3Department of Mechanical and Aerospace Engineering, Politecnico di Torino, 10129 Torino, Italy

**Keywords:** crack propagation, linear elastic fracture mechanics, stress intensity factor, vane

## Abstract

This work assesses the crack propagation at the most critical point of a second stage of a gas turbine blade by means of linear elastic fracture mechanics (LEFM). The most critical zone where the crack may nucleate, due to a combination of thermo-mechanical loads, is detected with an uncracked finite element (FE) model pre-analysis. Then the sub-modelling technique is used to obtain more precise results in terms of stresses within the area of interest. Simulations of the state of stress at the crack apex are performed through an FE model, using the Fracture Tool within ANSYS Workbench, and the stress intensity factors (SIFs) are determined accordingly. The Fracture Tool was previously verified on a simple model, and the results were compared with its analytical solution. Finally, the evaluation of the crack growth due to fatigue stress, creep, and oxidation is performed through in-house software called *Propagangui*. The crack behavior is estimated along with the component life. Results show an unexpected decrease in KI with increasing crack length and slowing of the crack growth rate with crack propagation. A detailed analysis of this behavior emphasizes that the redistribution of the stresses at the crack apex means that unstable propagation is not expected.

## 1. Introduction

### 1.1. Background

The continuous evolution of heavy-duty gas turbines (GTs), with the aim of improving machine performance, has resulted in the significant modification of its working conditions, particularly in terms of increasing temperatures and stresses. Especially regarding the expansion section of the engine, GT blades and vanes of the front stages, which are close to the combustion chamber, withstand extremely harsh operating conditions that trigger damage mechanisms [1]. These damage mechanisms, such as creep, fatigue [2,3], and oxidation, play an important role in crack initiation and propagation.

As a result, fracture mechanics (FM) is becoming increasingly important in the development and analysis of heavy GTs, not only to study possible crack behavior, but also to evaluate preliminary activities to avoid catastrophic failure. Some of these activities are selecting the required materials, defining the repair criteria, and formulating the component safe-life design.

Generally, stress intensity factor (SIF) evaluation is used to solve most of the engineering mechanical problems related to crack propagation, such as safety and service lifeestimation of cracked components. Both analytical and numerical methods can determine its main parameters. While analytical solutions can be applied to very idealized scenarios of geometry and stress distribution [4,5], numerical tools are more accurate when dealing with more complex geometry and boundary conditions. Recent software development has been achieved to simulate crack propagation behavior, especially in the framework of the finite element method (FEM), the boundary element method (BEM), the extended finite element method (XFEM), and meshless methods [6].

Commonly, complex structures with cracks are modeled through FEM because of its simplicity and versatility [7] using two main approaches. The implicit approach—also called smeared crack approaches—models the cracks using constitutive material models consistent with continuum damage mechanics (CDM) [8,9], while the explicit approach models the cracks through geometric entities. The explicit approach is suitable when dealing with structures with predefined crack paths. In this work, the explicit approach is used, as the crack growth direction is predefined by field feedback experience. On the one hand, working with FEM implies complex remeshing strategies as the crack evolves, especially under mixed mode conditions [10,11,12].

On the other hand, BEM and the dual boundary element method (DBEM)—a particular variant of the BEM—are characterized by a simplification of meshing processes, as the extension of the crack is discretized by adding new boundary elements, providing accurate gradients of the stress state at the crack apex [13,14]. Nevertheless, neither BEM nor DBEM are suitable for application to non-linear flow problems. In addition, the mathematics used under these approaches is unfamiliar to the engineering community, with matrices not easy to solve [15]. As shown in recent works, FEM and DBEM methods can work together to solve fracture problems in large structures or when faced with residual stresses caused by plastic deformations [16,17,18,19,20,21]. Generally, when a joint FEM–DBEM approach is used, FEM determines the global stress–strain fields, while a DBEM sub-model, obtained from the global FEM model, deals with the fracture problem.

The re-meshing issues from the FEM could be solved by the self-adaptive remeshing process. However, due to the complexity of applying and controlling any automated remeshing algorithm, the extended finite element method (XFEM) and meshless methods have been extensively and effectively applied to crack propagation. In XFEM [22,23,24], re-meshing is avoided by enriching the finite element interpolation functions using asymptotic expansions, which allows for an accurate representation of the crack propagation trajectories and singularity fields near the crack apex. Nevertheless, a priori knowledge of these enrichments is required, and complexities may arise in formulating finite elements and performing numerical integrations [25]. On the other hand, meshless methods [26,27] represent the domain by means of scattered nodes, where propagation will occur through the movement of computation nodes. These methods eliminate the re-meshing events but originate refined regions of nodes to account for arbitrary crack paths, thus requiring huge computational resources to perform the simulations.

### 1.2. Objective

The nature of the stresses present in the trailing edge of a vane is quite complex. Several effects act simultaneously:Thermal stresses due to thermal differences within the vane seats and large thermal differences between the internal cooling air and the external hot gas.Mounting stresses.Gas pressure.Local stress concentrations due to geometric discontinuities.

As the vanes operate well above the creep initiation temperature, all these stresses evolve over time, generally characterized by stress relaxation and redistribution.

In this scenario, the calculation of propagation is nontrivial. The crack propagates in a (mostly) displacement-controlled field in a multiaxial, non-proportional stress state. The general stress/strain/temperature state is significantly different from the experimental condition in which the propagation laws are generally calibrated. Therefore, it is important to improve the propagation calculation methodology to reduce the uncertainties associated with all the complexities mentioned above.

Despite the abundance of numerical methods to simulate crack propagation mechanisms, the joint contribution of fatigue, creep, and oxidation on crack growth is still an important topic that has not been thoroughly examined. Except for some theoretical investigations, there are no comprehensive numerical tools available to perform crack analysis in terms of these phenomena. In this work, a FEM approach—using ANSYS Code—was used to simulate the crack and to study fracture parameters, particularly the stress intensity factors (SIFs). Since the re-meshing process at each crack size is not avoided using this technique, the sub-modeling technique was applied, thus reducing the computational time. Furthermore, the crack propagation behavior in terms of fatigue, creep, and oxidation was studied using software developed in Ansaldo Energia, called *Propagangui*, which predicts crack propagation of cast *Ni*-base superalloys by identifying the propagation fraction caused by oxidation, creep, and fatigue crack growth [28].

Therefore, the main purpose of this work is to evaluate the propagation behavior of a crack located at the most critical points of a second stage GT vane for power generation (F-class). The behavior of a potential crack and the lifing of the component were estimated using linear elastic fracture mechanics (LEFM). The SIF was determined using FEM calculations, while the evaluation of crack growth, due to the joint phenomena of fatigue, creep, and oxidation, was carried out using the in-house software Propagangui

## 2. Numerical Model

The crack growth evaluation was conducted in three steps: (i) a preliminary analysis to obtain the most critical locations where the crack can nucleate because of thermo-mechanical loads, (ii) fracture parameter calculations using numerical tools to obtain crack dynamics through SIFs, and (iii) crack growth evaluation considering creep and oxidation. To estimate the component lifing, curves for different firing hours (FH) as a function of the number of shutdown (NS) were performed with different crack sizes. The FHs is the time (or number of cycles) that the vane can accomplish with stable crack propagation. For reasons of confidentiality, results were normalized by their maximum value. The flowchart of the procedure is detailed in Figure 1.

### 2.1. Preliminary Analysis

An FEM analysis was performed beforehand on the component to evaluate the stress state under the operating conditions and to obtain the critical areas where cracks may initiate. The component under investigation is a second stage vane on which a static analysis within the elastic field was developed with the software *ANSYS Workbench and ANSYS Mechanical APDL*. The FE model is depicted in Figure 2 and consists of a turbine vane with its turbine vane carrier sector (TVC) and the corresponding U-ring sector. For all these components, a refined mesh was prepared to improve the accuracy of the results.

The equivalent von Mises Stress at the GT vane and the displacement were attained. Furthermore, the two main damage mechanisms produced at the extreme working conditions—creep and thermo-mechanical fatigue—were also evaluated. Creep analysis was performed at different times according to maintenance intervals. The fatigue analysis was carried out with two different approaches, namely, considering or not the stress relaxation due to creep strain increase. In the first approach, no stress relaxation was considered, hence evaluating the number of cycles to crack initiation in the worst possible condition, R=σminσmax=0, whereas the second approach was performed considering the stress relaxation attained due to creep, which provides a higher number of cycles to crack initiation because this relaxation produces a shifted effect of the load cycle mean stress.

Figure 3 and Figure 4 show the boundary conditions, normalized to the maximum value, in terms of mechanical loads and temperature distribution. Figure 3 illustrates the pressure distribution along the components such as external airfoil, inner and outer platforms, internal cavities, etc.

Figure 4 shows the (normalized) temperature distribution in the vane. It depends mainly on the gas heat transfer coefficient, the material thermal conductivity, and the internal cooling system.

### 2.2. Fracture Parameters Assessment

In this work, the crack propagation behavior was evaluated by means of the SIF. The complex geometry to be analyzed in addition with the extreme conditions that the component must withstand meant that the calculations of the fracture parameters were performed through numerical tools, in particular by means of the *Fracture Tool* in *ANSYS Workbench*. The different calculation steps are detailed below.

#### 2.2.1. Sub-Modeling

The most critical section was identified, with the preliminary analysis, at the platform exterior trailing edge (PETE). To reduce computational times and obtain more accurate results, the sub-modeling [29] technique was applied to the uncracked vane. According to this technique, the displacement field and the temperature gradient obtained from a coarse model were applied to the sub-model as boundary conditions, obtaining an accurate and highly refined response at the interested area. The remaining boundary conditions, such as mechanical loads, were applied directly to the sub-model. The steps of the sub-modeling [29] technique are listed below.
The solution of the full coarse model with *ANSYS Mechanical APDL* provides the displacement field and temperature distribution.The main region of interest is finely modelled. The coarse model is cut as illustrated in Figure 5. The sub-model faces that are “inside” the full model are called cut boundary faces.In the *ANSYS Workbench* project, the sub-model task is attached to the engineering data and to the full model solutions, as shown in Figure 6.The following loads are applied to the-sub model:
○Imported cut boundary conditions (displacements) from the full model.○Interpolated body temperature from the full model.○Mechanical loads (pressure distribution) directly applied on the sub-model.

#### 2.2.2. Meshing

Furthermore, with the aim of guaranteeing the approximation of all the singularities within the PETE and obtaining a higher stress gradient, a sensitivity analysis of the mesh was developed. This analysis was done with the SIF evaluation, adopting a mesh size that gave a mesh-independent status. The implemented mesh had a sphere of influence of 8 mm in radius and 0.5 mm in element size, as illustrated in Figure 7.

The rest of the sub-model was meshed with *quadratic tetrahedron elements*, such as the element SOLID187, with dimensions of 1.5 mm. This element is a higher order 3-D, 10-node element with a quadratic displacement behavior, well suited to modelling irregular meshes. The element geometry is illustrated in Figure 8. Figure 9a shows the refined mesh, while Figure 9b details the sphere of influence that give a more refined mesh around the critical location.

#### 2.2.3. Crack Modeling

Among the several techniques employed to simulate fracture behavior of complex structures, the finite element method (FEM) is largely adopted. This work used the *Fracture Tool* within *ANSYS Mechanical Workbench,* in particular the *Arbitrary Crack Method*, to define the crack and to compute the fracture parameters.

The applied method involves simulating the crack through a surface body. In this case, a circular shape was chosen, putting it on the most critical zone (PETE), as shown in Figure 10. Therefore, by increasing or diminishing the radius of the circular shape, the crack length is increased or decreased. Furthermore, a new coordinate frame was needed to define the crack propagation direction. From field feedback experience, the preferred crack propagation direction was obtained, setting the origin in the crack apex, the X-axis towards the crack propagation direction, and the Y-axis towards the crack opening direction, as illustrated in Figure 11.

Furthermore, within the Fracture Tool, the type of crack method and its correspondent properties must be specified. For example, one needs to specify the already-created coordinate frame, the crack surface, the mesh method set to tetrahedrons, and the number of mesh contours for the crack shapes, set to six.

Once all these parameters are set, the model is ready to generate the fracture meshing. This is a post mesh process, which occurs in a separate step after that the base mesh is generated, overriding its settings within the crack limits, permitting the crack front analysis and therefore making it ready for the SIF calculations.

## 3. Calculation of the Stress Intensity Factor

The stress and deformation status around the crack apex are not enough to assess the catastrophic failure of a structure. The computation of fracture parameters and its comparison with the material fracture toughness must be performed. The SIFs are computed along the crack front through the *Fracture Tool*, using the interaction integral method, which is defined as
(1)$$I_{0}=-\frac{\int_{V}^{\;}q_{i,j}\left[\sigma_{kl}\varepsilon_{kl}^{aux}u_{k,i}-\sigma_{kj}u_{k,i}^{aux}\right]dV}{\int_{S}^{\;}\delta q_{n}dS}\;$$
where $\sigma_{ij}$, $\varepsilon_{ij}$, and $u_{i}$ are stress, strain, and displacement, respectively; $\sigma_{ij}^{aux},\;\varepsilon_{ij}^{aux},\;\mathrm{and}\;\;u_{i}^{aux}$ are stress, strain, and displacement of the auxiliary field, respectively, while $q_{i}$ is the crack extension vector. Moreover, if the thermal and initial strains exist in the structure, and the surface tensile acts on crack faces, the interaction integral is expressed as
(2)$$I=I_{0}+\frac{\int_{V}^{\;}\left[\sigma_{kl}^{aux}\varepsilon_{kl,i}^{th}-\sigma_{kl}^{aux}\varepsilon_{kl}^{0}\right]q_{i}dV}{\int_{S}^{\;}\delta q_{n}dS}-\frac{\int_{S}^{\;}\left[t_{k}u_{k,i}^{aux}\right]q_{i}dS}{\int_{S}^{\;}\delta q_{n}dS}\;$$
where $\varepsilon_{ij}^{th}$, $\varepsilon_{ij}^{0}$, and $t_{i}$ are thermal strain, initial strain, and tensile on crack surfaces, respectively. As a result, the interaction integral method is associated with the stress-intensity factors
(3)$$I=\frac{2}{E^{*}}\left(K_{1}K_{1}^{aux}+K_{2}K_{2}^{aux}\right)+\frac{1}{\mu }K_{3}K_{3}^{aux}\;$$
where
$K_{i}\;\left(i=1,\;2,\;3\right)$ are Mode I, II, and III stress-intensity factors,$K_{i}^{aux}\left(i=1,\;2,\;3\right)$ are the auxiliary Mode I, II, and III stress-intensity factors,E is Young’s modulus, ν is Poisson’s ratio, and μ is the shear modulus, and$E^{*}=E$ for plane stress or $E^{*}=E/\left(1-\nu^{2}\right)$ for plane strain.

Additionally, to guarantee the accuracy of the stress-intensity factor calculations, the local crack apex coordinate system must fulfill the following specifications:Local X-axis pointed toward the crack propagation.Local Y-axis pointed toward the normal crack surface (opening crack direction).Local Z-axis pointed toward the tangential direction of the crack front.

The local coordinate frame consistency across all the nodes along the crack front is important to obtain the path-dependency, which results in a correct SIF behavior. Additionally, it can be highlighted that the interaction integral method applies area integration for 2-D problems and volume integration for 3-D problems.

However, prior to evaluating the SIFs at the critical vane location, a fracture parameter analysis on a test plate was performed as a validation test of the procedure. The test plate, modelled with *ANSYS SpaceClaim*, is represented in Figure 12a. A crack with a rectangular shape was inserted, following the procedure explained in the previous subsection, and the plate was loaded with a normal tensile load of 1000 N at the top surface and constrained at the opposite surface, as represented in Figure 12b.

Moreover, the base mesh was set to tetrahedrons, refining the crack region. The main objective of this analysis was to compare the numerical values of $K_{I}$ given by ANSYS with their theoretical value
(4)$$K_{I}=Y\sigma \sqrt{\pi a}$$
as a function of the crack depth *a* as reported in [30]. The increase of *a* was performed by increasing the rectangular length at each time (1, 1.5, 3, 6, and 10 mm). The same procedure was followed to evaluate the SIFs at the turbine vane. The crack length was increased by incrementing the radius of the circular shape and computing for each length the SIFs. The crack lengths studied were 1, 2.5, 3, 5, 7.5, and 10 mm.

Nevertheless, the SIFs were computed by applying the interaction integral method at each node found along the crack front in six different path contours. To study all these values, it was decided to calculate the mean value of the five-last contours—the more reliable ones—which gave the average $\bar{K_{I}}$ found at each node along the crack front. However, to obtain a clear evolution of $K_{I}$ with length *a*, the average value of the $\bar{K_{I}\;}$ found along the crack front was evaluated.

### Crack Growth Assessment

The crack growth behavior due to fatigue, creep, and oxidation phenomena was studied through an in-house tool called *Propagangui*, provided from Ansaldo Energia company. It is software developed in the MATLAB environment, which performs life evaluation of cracked components. Particularly, it carries out crack propagation calculations, failure assessment, and probabilistic failure evaluation using Monte Carlo analysis. In this work, the two first approaches were used to analyze crack growth. Crack propagation is based on the LEFM theory considering fatigue, creep, and oxidation as the main processes for crack growth [31]. The different contributions are modelled with the equations reported below.

**Fatigue Crack Growth** modelled through Paris’ law equation:


(5)
$$\frac{da}{dN}=C\Delta K^{n}\;$$


**Creep Crack Growth** modelled through Arrhenius’ dependent equation:


(6)
$$\frac{da}{dt}=A_{0}K_{MAX}^{m}e^{-\frac{Q}{RT}}\;$$


**Oxidation** modelled through a sub-parabolic Arrhenius-dependent equation, which relates depletion zone thickness and time:


(7)
$$d{\gamma^{\prime }}^{n}=A_{0,\gamma^{\prime }}e^{\frac{Q_{\gamma^{\prime }}}{RT}}t\;$$


The oxidation contribution is calculated assuming that the $\gamma^{\prime }$-depleted zone corresponds to a crack of the same length; therefore, the Oxidation Crack Growth is modelled as
(8)$$\frac{da}{dN_{OXI}}=\frac{d\sum_{T_{i}}^{\;}\frac{\Delta a_{oxi}}{\Delta t}T_{i}}{dN}t_{i}\;$$

The model calculates the contribution of these phenomena separately for a given operating cycle and assumes superposition of effects,
(9)dadNtot=dadNOXI+dadNCREEP+dadNFATIGUE,
which is the most common assumption in industrial applications. It should be outlined that a critical temperature of the material was defined. Below this temperature, the crack growth was mainly due to fatigue loads, because of the absence of $\gamma^{\prime }$ depletion zone and creep effect.

*Propagangui* evaluates the failure of a component with a crack using the failure assessment diagram (FAD), reported in Figure 13 as an example. International standards and codes of practice [32,33] recommend slightly different implementations of the failure assessment line. Additionally, different calculation options are available depending on the type of input available and accuracy required. Figure 13 illustrates how the assessment result is affected by these choices. Its main objective is to assess whether the component will fail when it crosses the FAD line or maintain a stable behavior. The X-axis reports the variable $L_{r}=\sigma_{ref}/\sigma_{y}$ $\left(0\leq L_{r}\leq L_{r,max}\right),$ where an exceedance of its limit implies plastic collapse. The yield stress $\sigma_{y}$ is also referred to as $R_{P02}$.

The Y-axis represents the variable *Kr*, which is a function that does not depend on the load or geometry but only on the deformation behavior of the material, and its limit will ensure that the unstable propagation condition is not reached. Moreover, the FAD can be established in different ways depending on the available inputs and the level of safety needed, where more inputs will lead to a less conservative diagram and vice versa.

*Propagangui* evaluates the assessment point for each cycle, and thus the crack propagation behavior, if exceeding the FAD limit, implies failure of the component. As can be expected from the previous statement, this failure can be due to plastic collapse, if the assessment point reaches the right boundary of the FAD, or due to unstable propagation, if the assessment point exceeds the upper boundary of the FAD. Additionally, the reached assessment point is expressed by the following coordinates:(10)$$\{\begin{array}{c} x=Lr=\frac{\sigma_{ref}}{\sigma_{y}}\\ y=Kr=\frac{K_{eq}}{K_{mat}}=\frac{K_{I}}{K_{Ic}} \end{array}\;$$
where $\sigma_{ref},$ also referred to as $\sigma_{NetSection},$ is the von Mises equivalent stress of the uncracked component found at the corresponding crack depth, considering only primary stress, and $\sigma_{y}=R_{P0.2}$ is the offset yield strength of the material. On the other hand, the $K_{eq}$ can be equal $K_{I},$ or a function of $K_{I}$, $K_{II}$, and $K_{III}$, depending on the multiaxiality state in the region of the flaw, while the $K_{mat}$ is the material fracture toughness ($K_{IC}$). Then the software will make an estimation of the propagation analysis providing the listed graphs:Crack evolution plot, which reports the crack growth among cycles.Propagation plot, which illustrates the crack propagation rate in the stress intensity factor range.FAD plot, in which the assessment point is represented.

The propagation calculation will stop if one of the following conditions is satisfied:The assessment point meets the FAD line where plastic collapse or unstable propagation can take place.The crack grows until the specified number of cycles without reaching the FAD line, considering in simulated conditions the component as safe.The calculation achieves the last crack length for which the SIF was obtained (in this case it is 10 mm).To proceed with the assessment of the vane, the following inputs required to carry out the analysis are recollected from ANSYS.The set of SIFs already obtained in the previous subsection, where the use of multiaxiality correction—$K_{I}$, $K_{II}$, and $K_{III}$—can be emphasized, since the three are of the same order of magnitude.The temperature along the crack path of the uncracked component to apply the oxidation and creep phenomena whether the temperature is above the critical one.The von Mises equivalent stress along the crack path of the uncracked component to establish the assessment point. It is important to outline that the stress related to the assessment point considers only the primary stress. Therefore, it multiplies the obtained stress by the ratio between the primary and total stresses. To obtain this value, three different analyses were performed on ANSYS: with all loads, with mechanical loads only, and with thermal loads only.The net area of the crack is defined as (total area−crack area)/total area.

Additionally, the inputs related to the material (critical temperature, fracture toughness, yield strength, etc.) and the parameters related to fracture mechanics (oxidation, creep, and fatigue coefficients) should also be introduced. Moreover, an initial crack size is given (in this analysis it was 1 mm) and a cyclic target (3000 cycles) which, as it was mentioned before, can be one of the causes of the end of the calculation. Furthermore, to evaluate the crack propagation behavior, an operating cycle of 10 firing hours (FH) was given to the program.

## 4. Results

This section details the results of the various analyses developed in this work.

### 4.1. Preliminary Analysis

Displacements from static analysis, normalized to their maximum value, are depicted in Figure 14. Results were consistent in that an outer radial increase in the vane could be seen following the radial expansion of the TVC.

The stress evaluation in terms of the equivalent von Mises, normalized to the maximum value, emphasized that the yield strength of the material was exceeded in two regions, namely, PETE and the platform internal trailing edge (PITE), as shown in Figure 15. Therefore, these two areas were considered to be critical. Consequently, the two life-limiting phenomena presented below, creep and fatigue, focused on these life-limiting areas.

The creep analysis emphasized that the creep limit of the material was exceeded after a few firing hours, and the creep verification criteria were not fulfilled, as illustrated in Figure 16, where the creep strains were normalized with the material creep limit. This result highlighted the criticality of the area, whereas the absolute strain values, affected by the local geometry, mesh and boundary conditions, must always be interpreted. Moreover, the PETE is always the most critical area. Figure 17a shows the creep analysis performed at 66 kFH, with results normalized to the material creep limit value. Areas where the creep limits were exceeded are highlighted. The fatigue (LCF) assessment is shown in Figure 17b, and it shows that the required number of cycles was not reached in the PETE for either of the two approaches performed, namely, considering or not the stress relaxation due to the creep strain increase. However, it should be remarked that from the first approach, considering stress relaxation, the number of cycles to crack initiation was lower than the second approach, as expected. Therefore, from this preliminary analysis, it can be concluded that the PETE is the most critical part of the component from which a crack may nucleate.

### 4.2. Fracture Parameters Assessment Results

#### 4.2.1. Sub-Modeling

To verify the correctness of the sub-modeling technique, a comparison was performed between the equivalent von Mises stress found in the full model and the sub-model. Figure 18 shows the comparison. The left part of Figure 18 shows the whole vane stress analysis, cropped to the interested area, while the right part shows the sub-model with its boundary conditions. The comparison assessed the correctness of the sub-model implementation, since the state of stress was almost the same in the two models.

#### 4.2.2. Stress Intensity Factor Results

The analysis performed on the test plate showed an increase of the $K_{I}$ with the crack length a, as reported in Figure 19. The comparison between the analytical results and the computation of the SIFs by the *Fracture Tool* within *ANSYS* validated the procedure. The initial crack size was arbitrarily defined with a depth of 1 mm. This choice was due to several considerations:The sensitivity of nondestructive testing (NDT) techniques generally used in the energy industry: although several techniques can detect defects and cracks smaller than 1 mm, this length is a reasonably safe assumption.The crack initiation process in a high-load area generates cracks that can propagate into the component. The transition between the crack initiation phase and subsequent propagation is a gray area. Both field findings and specimen analysis show that the initial crack size after initiation is significantly less than 1 mm, but it cannot be ruled out that under some conditions, the initiated crack may reach a length of 1 mm in a very limited number of cycles. Therefore, the assumption that the crack after initiation is 1 mm deep is a reasonably safe choice.

The target number of cycles for the calculation was set at 3000 cycles, which corresponds to more than 8 years in a daily cycle operating profile. Results obtained on the vane, at the PETE area, are depicted in Figure 20 and Figure 21. Figure 20 shows the evolution of $K_{I}$ along the crack front for different crack lengths, normalized with the material fracture toughness. The increase of $K_{I}$ as it reached the pressure side of the airfoil is remarkable. Figure 21 depicts in more detail the unexpected trend of $K_{I}$ with increasing crack length a, normalized with the material fracture toughness. The main result is an unexpected reduction of the stress intensity factor $K_{I}$ with increasing crack length a, which additionally is much smaller than the fracture toughness $K_{c}$. This unexpected result can be explained with the following considerations. The nature of the stresses acting on the trailing edge of a vane is quite complex. The dominant part is a stress state generated by temperature differences between the different positions of the cooled vane (e.g., the inner surface, in contact with the cooling air, and the outer surface, in contact with the hot gas), plus some thermal stresses generated by the different temperatures between the vane profile, the disc, and the inner ring, plus some pressure generated by the exhaust gas flow, plus some contribution of concentration factors related to geometric discontinuities, plus residual stresses from the vane assembly. This leads to a non-proportional, multiaxial stress state that tends to evolve after the effect of viscous phenomena, generally giving rise to stress relaxation and redistribution. In addition, strong thermal gradients generate strong stress/deformation gradients. The stress state can change drastically within millimeters. In addition, the presence of the crack itself causes stress redistribution. In such a complex scenario, it is quite understandable that the stress intensity factor now depends on the square root of the crack size, as predicted by theory, and can easily decrease as it is propagating toward the least stressed region of the vane. Performing an FE-based analysis, such as the one we conducted in this paper, is precisely intended to highlight these phenomena and these “unexpected” dependencies.

The slowing of the crack growth rate leads to the conclusion that unstable propagation is not expected.

Moreover, the equivalent von Mises stress at PETE for each crack length is represented in Figure 22. Figure 22a shows the equivalent stress at the pressure side of the airfoil, where a redistribution of stress at the crack apex can be seen. Additionally, from Figure 22b, a lower stress is predicted at the crack apex, confirming that $K_{I}$ at the suction side is lower than that on the pressure side.

Furthermore, the two other SIFs, $K_{II}$ and $K_{III}$, were also determined, and the results are shown in Figure 23, normalized with the material fracture toughness. Both $K_{II}$ and $K_{III}$ were lower than $K_{I}$, but of the same order of magnitude. Additionally, from the comparison with the material fracture toughness, it is also important to note that unstable crack propagation was not expected from this first analysis, since the fracture toughness was well above the SIFs obtained.

### 4.3. Crack Growth Assessment Results

The calculation of crack propagation requires one to make several assumptions about the crack characteristics. Different assumptions are generally made depending on the type of calculation that is generally performed. For example, handling a manufacturing nonconformity requires a high degree of accuracy in the type of crack being modeled; on the other hand, propagation calculations that are required to define the desired quality of production (evaluation criteria) assume the existence of “possible defects” with unknown orientation and direction of propagation. Consequently, in industry, each calculation requires a different set of assumptions to describe the crack, its aspect ratio (the ratio of its depth to its surface extent), and its evolution.

One of the most important aspects when discussing the use of fracture mechanics in an industrial scenario concerns the time required to set up the calculation and the resulting computational resources needed to perform it. The highest accuracy is achieved by explicit modeling of the crack at the CAD level, with subsequent evaluation of the stress intensity factor, SIF, using the finite element solver. This method can account for the local redistribution of energy generated by the presence of the crack. Unfortunately, this method is extremely challenging, because the modeled crack has a certain depth/extension, and after its propagation, the finite element calculation must be repeated. This expensive procedure can be performed sporadically for some special cases (such as the one shown in this article), but it cannot be used extensively for all Ansaldo fracture mechanics calculations. Consequently, the results of these expensive exercises should be used to improve the calculation methodology based on simpler and faster methods. The possibility of performing fast propagation calculations with higher accuracy based on results obtained in the calculations shown in this article is the main motivation and the most important expected result.

This section reports the results obtained with the simulation performed with the code *Propagangui*. Figure 24 shows the crack growth evolution with the number of cycles, considering the equivalence 1 cycle = 10 firing hours. The software stops the calculation when the crack length reaches 10 mm, which is the last crack length for which the SIF was obtained. The slope of this graph represents the crack growth rate. The slowdown as the crack evolves is evident, meaning that the rate of crack propagation decreased.

On the other hand, Figure 25 shows the crack propagation rate as a function of SIFs. This plot emphasizes the linearity of the fatigue and creep propagation with SIFs, as expected from Equations (5) and (6), and the non-dependence of the oxidation contribution. Figure 26 shows the failure assessment diagram (FAD). The evolution of the assessment point at different cycles was within the limits. Therefore, neither plastic collapse, nor unstable propagation was expected.

### 4.4. FH–NS Diagram Results

Figure 27 depicts the evolution of the cracks at the different operating cycles. This graph gives the number of cycles needed to reach the expected crack length, namely 2, 3, 5, and 10 mm. Data represented in Figure 27 can be rearranged to create the FH–NS diagram, where FH are the firing hours, and NS (normal shut down) are the cycles. It is also worth noting the different rate of crack growth at each operating cycle. The number of cycles to reach the limiting crack length (10 mm) decreases with increasing firing hours per cycle. This result is because with a higher number of hours per cycle, the total number of hours to reach limiting crack length (as shown) is much higher, and therefore, the phenomena of fatigue, creep and oxidation count much more, arriving to the limiting crack length earlier.

### 4.5. FH–NS Diagram Approach

Another diagram, the FH–NS diagram, was also derived to estimate component lifing. NS–FH lines for different “critical” crack sizes could be extracted from the software *Propagangui* and then the number of cycles or firing hours needed to reach the critical crack size. Several operating cycles with different firing hours for each operating cycle were analyzed. The aim of this analysis was to obtain the number of cycles to reach the target crack length (2 mm, 3 mm, 5 mm, and 10 mm) for the different operating conditions. The relationship between equivalent firing hours of each cycle and the total number of cycles in terms of number of start-ups gave the total number of equivalent operating hours (EOH) and therefore the requested FH–NS diagrams. This diagram is presented in Figure 28. This plot shows the different lines below which the vane could withstand the external load without reaching the “critical” crack size. Firing hours and number of cycles were normalized to the values to reach the first maintenance interval of the component.

## 5. Discussion

The results presented in this analysis show unexpected behaviors of SIF $K_{I}.$ From a theoretical point of view, the increase of $K_{I}$ with the increase of a is expected. However, in this work the reduction of $K_{I}$ with increasing crack length after a certain crack length value was achieved. This result can be explained with the complex geometry of the vane in addition to the extreme working conditions that allow a stress redistribution at the crack front apex. This stress redistribution was predicted by the simulations. Therefore, due to this phenomenon, no unstable amplification of the stresses along the crack front is expected.

Furthermore, the increase of $K_{I}$ along the crack front as it reaches the pressure side was predictable, since from the preliminary analysis done, the higher stress state was reached in this side of the airfoil. Therefore, the amplification of the stresses that the crack produces is expected to be bigger in the pressure side.

The SIF $K_{I}$ better describes the crack propagation dynamics. Nevertheless, the other two SIFs, $K_{II}$ and $K_{III}$, were used in the multiaxiality correction to perform the crack growth study in terms of fatigue, creep, and oxidation, as they should not been neglected because they are of the same order of magnitude.

The crack growth analysis also shows a reduction of the crack growth rate as it evolves. This decrease is again explained with the redistribution of stresses at the crack apex. The reduction of the stress achieved as the crack evolves gives no unstable propagation.

From the FAD diagram, SIF evaluation, and comparison with the fracture toughness of the material, neither unstable propagation nor plastic collapse is expected. In fact, from the FH–NS diagram, the number of cycles or firing hours until the arrival of a certain crack size was obtained, thus estimating the lifing of the component. Finally, the FH–NS diagram confirms the slowing of the crack growth rate because the crack needs more time, in terms of firing hours or number of cycles, to propagate.

## 6. Conclusions

To set the framework for this work, it is worth noting that gas turbine blades are characterized by significant thermal stresses, which, being self-limiting stresses controlled by displacement, tend to have significant redistribution in the presence of material failure or crack formation. This is one reason why detailed modeling of the crack and the corresponding stress intensity factor at different crack lengths is important. This paper presented a crack assessment on a second-stage gas turbine vane via the LEFM approach, in particular by means of SIF evaluation in addition to crack propagation analysis in terms of fatigue, creep, and oxidation. The crack growth behavior was studied, under a potential crack propagation hypothesis, on the most critical location of the GT vane—PETE—where a crack may nucleate due to a combination of thermo-mechanical loads. The expected result is to determine whether crack propagation exhibits stable behavior or propagates in an unstable manner, causing possible component failure.

The main conclusions obtained from the analysis performed in this work are listed below:The PETE is the most critical zone where the crack may nucleate due to thermo-mechanical loads.The reduction of the SIF $K_{I}$ with increasing crack length a is explained by the redistribution of stresses that occurs at the crack apex.The $K_{I}$ increases along the crack front as it reaches the pressure side.The higher value of $K_{I}$ compared to $K_{II}$ and $K_{III}$ means that $K_{I}$ better describes the crack propagation dynamics.Material fracture toughness is always above the SIF.Crack growth rate slows down as the crack length increases.The assessment points are within the limits in the FAD diagram.The estimated crack size curves on the FH–NS diagram emphasize the slowing of the crack propagation.

It can be concluded that the stress redistribution at the crack apex gives a decreasing crack growth rate, and the crack propagation is stable.

The obtained results represent an important milestone for Ansaldo Energia, since they are expected to facilitate future propagation calculations at the second vane trailing edge. Further improvements of the calculation and the method might focus on the investigation of elastic–plastic fracture mechanics and/or an improved approach to account for the creep crack growth contribution.

Furthermore, based on FM calculations and qualifications of the repairing process according to the company technical specification, the repaired components are again assembled into the engine, and a further maintenance interval is guaranteed. Moreover, as further risk mitigation action to assess the lifing of the components, minor inspections within the standard maintenance interval are scheduled by means of borescope investigation.

## Figures and Tables

**Figure 1 materials-15-04694-f001:**
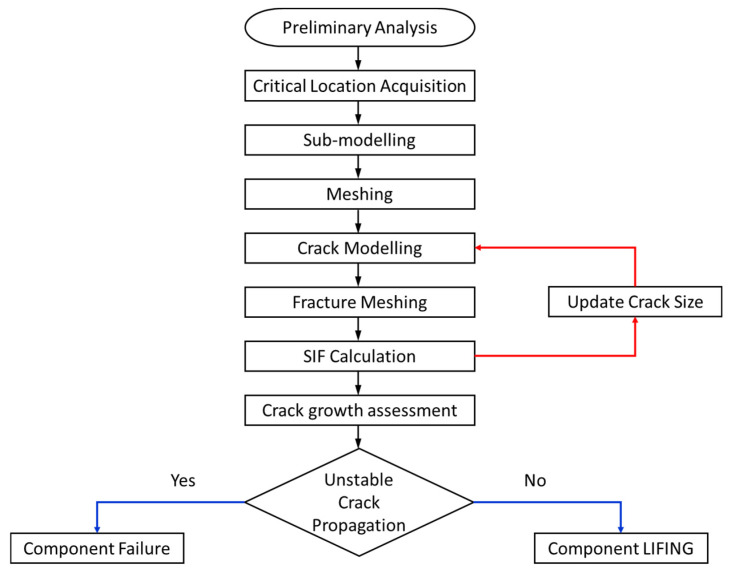
Flowchart of the whole crack growth procedure on the studied component.

**Figure 2 materials-15-04694-f002:**
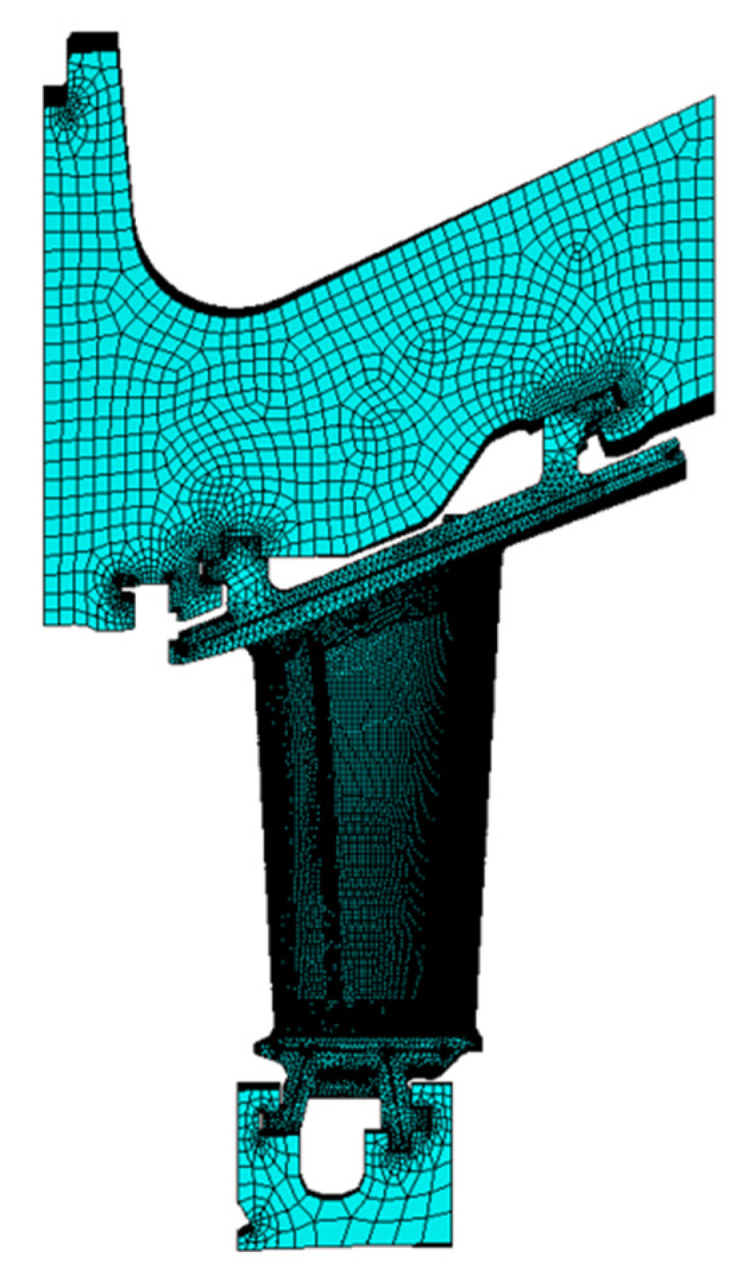
Finite element (FE) model of a turbine vane with its turbine cane carrier (TVC) and U-ring sector.

**Figure 3 materials-15-04694-f003:**
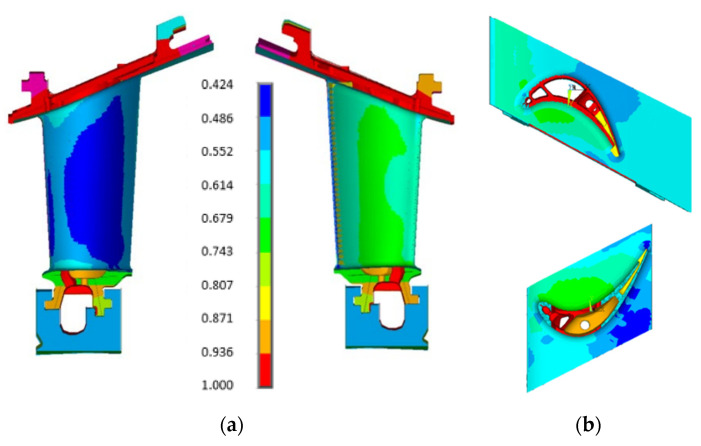
Mechanical load distribution on the turbine vane: (**a**) pressure distribution along the external airfoil; (**b**) pressure distribution along the outer and inner platforms.

**Figure 4 materials-15-04694-f004:**
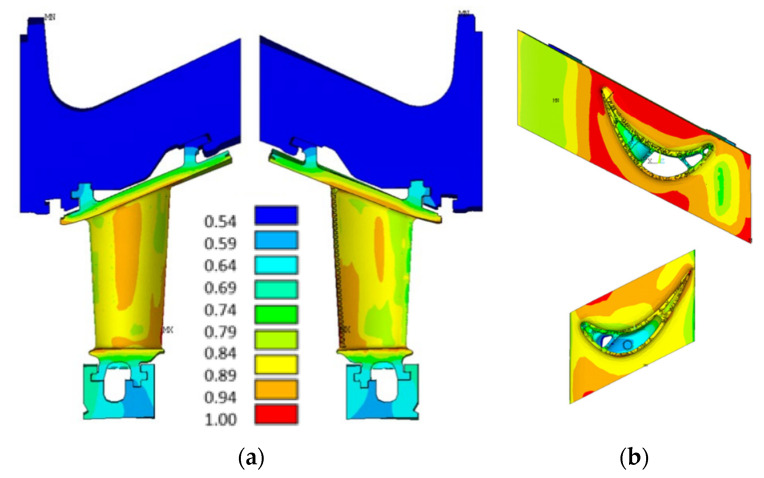
Temperature distribution along the turbine vane: (**a**) temperature distribution along the airfoil; (**b**) temperature distribution along the outer and inner platforms.

**Figure 5 materials-15-04694-f005:**
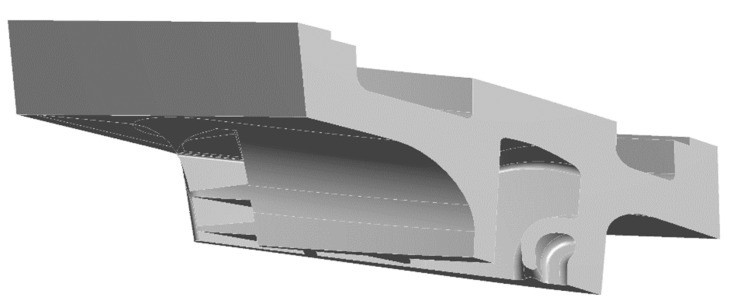
Sub-model of the vane around the platform external trailing edge (PETE) region.

**Figure 6 materials-15-04694-f006:**
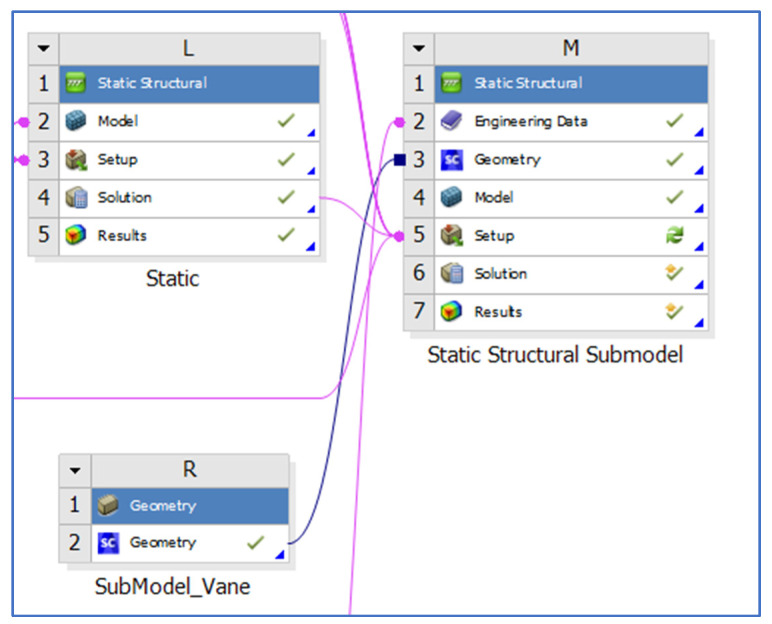
Sub-model attached to the engineering data and solution of the full model in ANSYS Workbench.

**Figure 7 materials-15-04694-f007:**
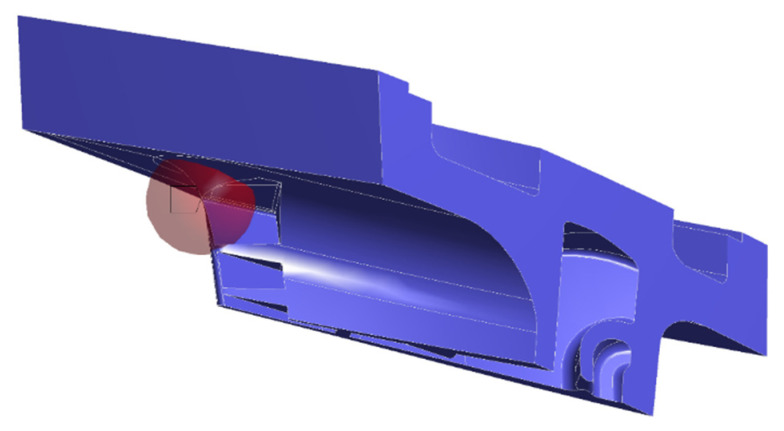
Mesh around the PETE with a sphere of influence of 8 mm radius and 0.5 mm element size.

**Figure 8 materials-15-04694-f008:**
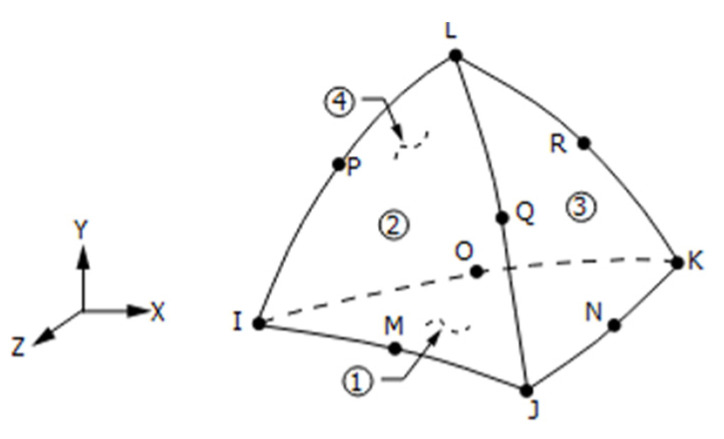
SOLID 187 geometry.

**Figure 9 materials-15-04694-f009:**
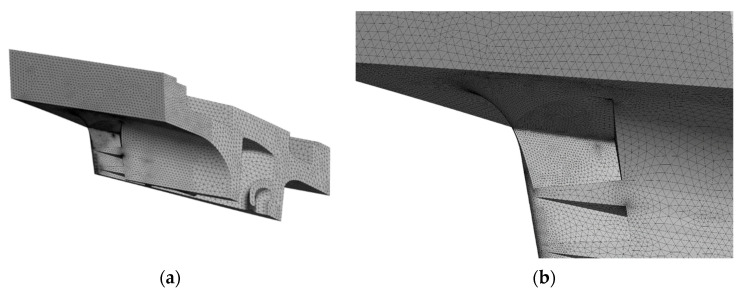
Mesh results: (**a**) refined mesh on sub-model; (**b**) mesh result around the PETE due to the sphere of influence.

**Figure 10 materials-15-04694-f010:**
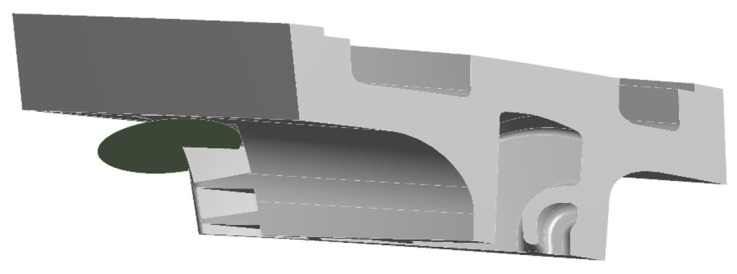
Crack modeling through a circular surface at the PETE.

**Figure 11 materials-15-04694-f011:**
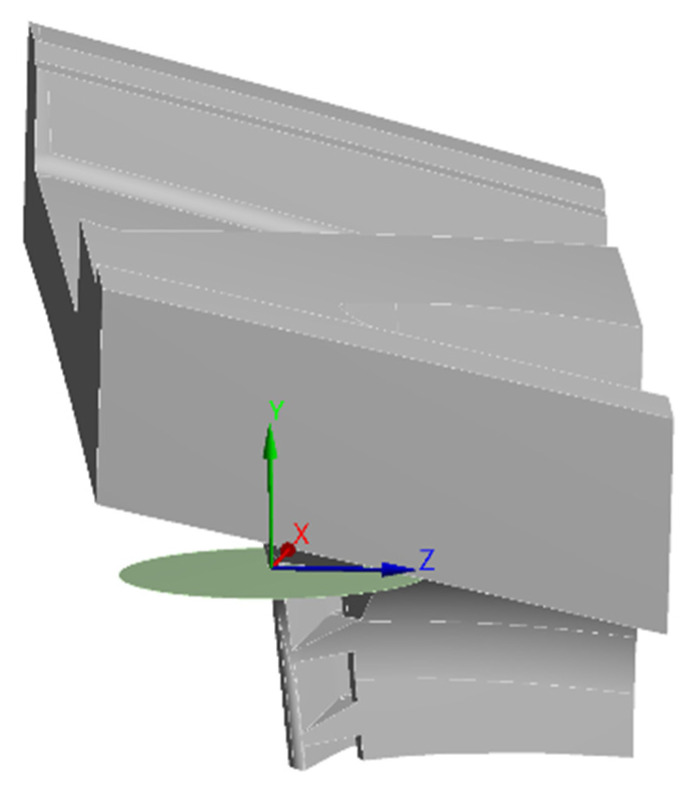
Coordinate system at the crack apex.

**Figure 12 materials-15-04694-f012:**
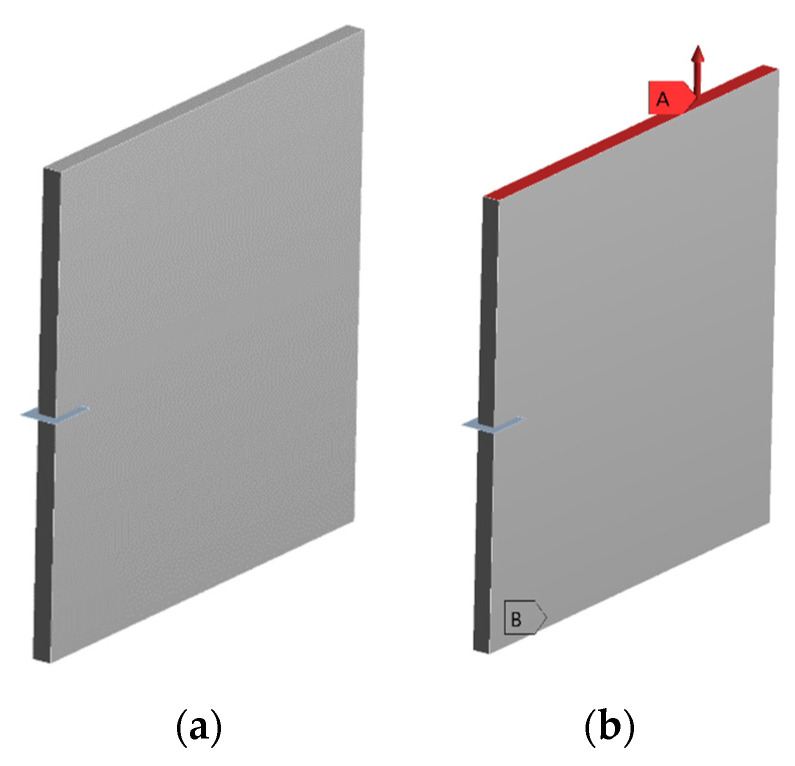
Test plate: (**a**) test plate with a rectangular crack shape in the middle of the specimen; (**b**) test plate with the normal force applied on the top surface (A) and constrained at the opposite surface (B).

**Figure 13 materials-15-04694-f013:**
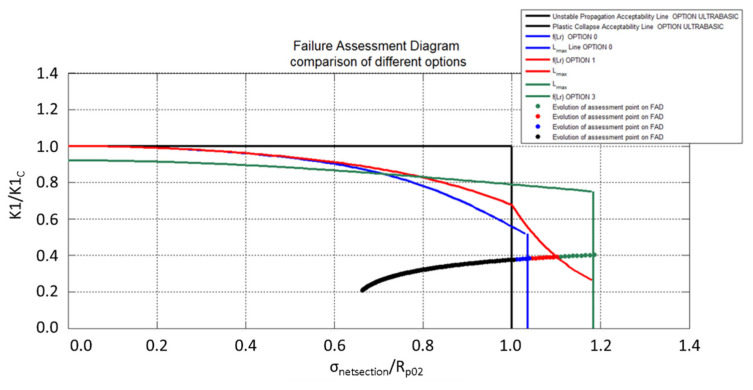
Example of a failure assessment diagram (FAD).

**Figure 14 materials-15-04694-f014:**
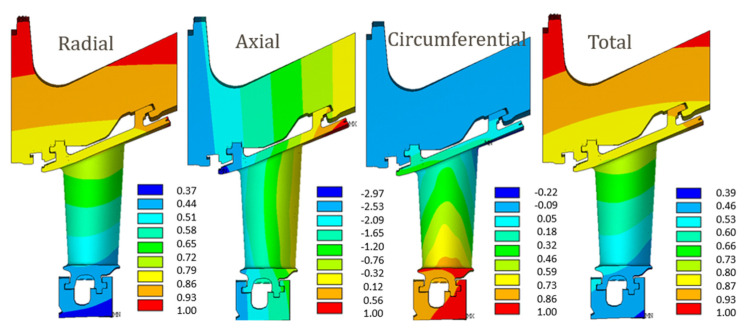
Distribution of the normalized displacements (with respect to the maximum displacement) on the vane.

**Figure 15 materials-15-04694-f015:**
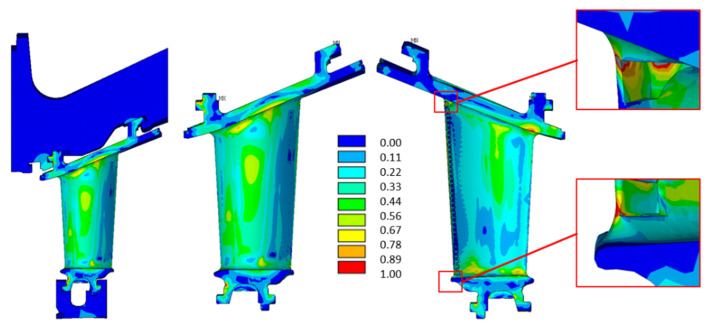
Equivalent von Mises stress. The yield strength of the material is exceeded in the two areas marked in the images, the platform exterior trailing edge (PETE) and the platform interior trailing edge (PITE). Stresses are normalized with respect to the maximum von Mises stress value.

**Figure 16 materials-15-04694-f016:**
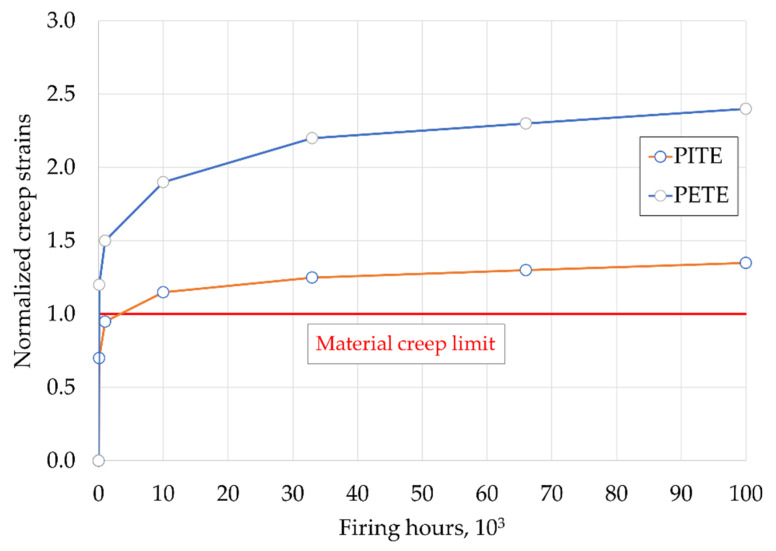
Normalized creep strains vs. firing hours at the PETE and PITE.

**Figure 17 materials-15-04694-f017:**
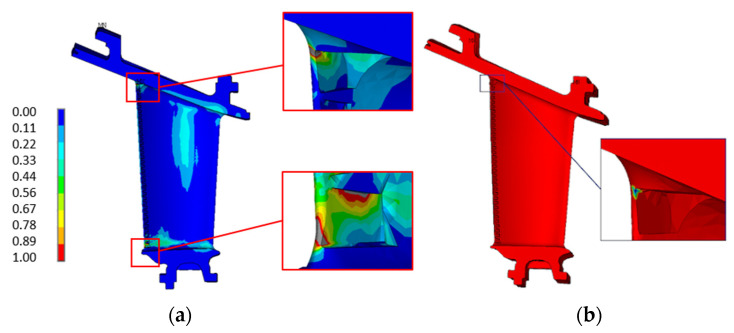
Damage analysis results. (**a**) Creep analysis at 66 kFH, where the creep limit is exceeded in critical areas; (**b**) LCF analysis with no stress relaxation.

**Figure 18 materials-15-04694-f018:**
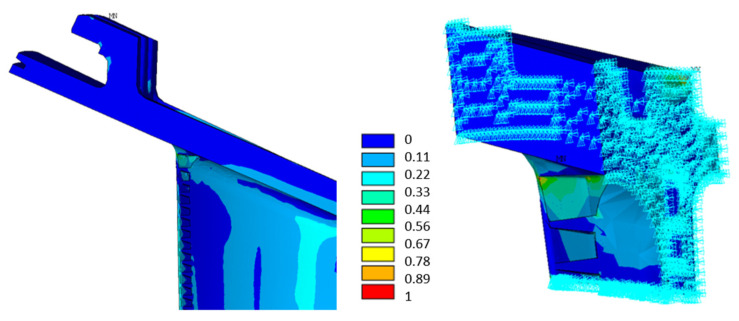
Comparison between the full model (on the **left**) and the sub-model (on the **right**) equivalent Von Mises stress.

**Figure 19 materials-15-04694-f019:**
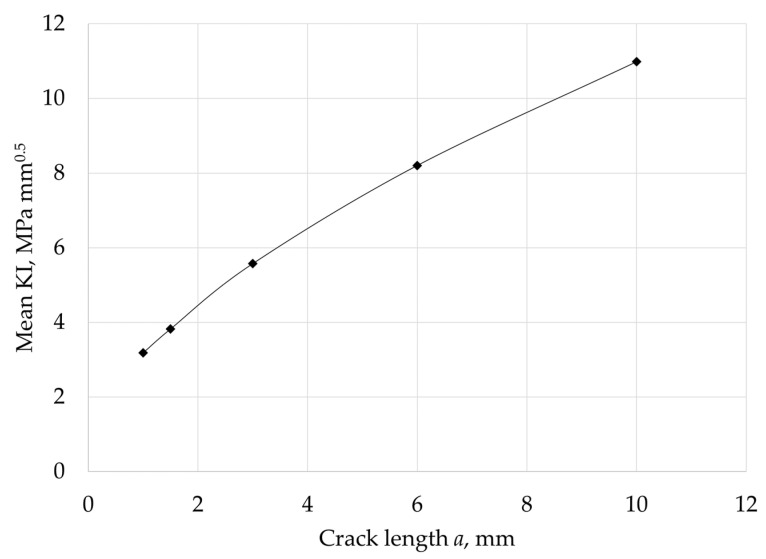
Evolution of $K_{I}$ with the crack length a of the test plate.

**Figure 20 materials-15-04694-f020:**
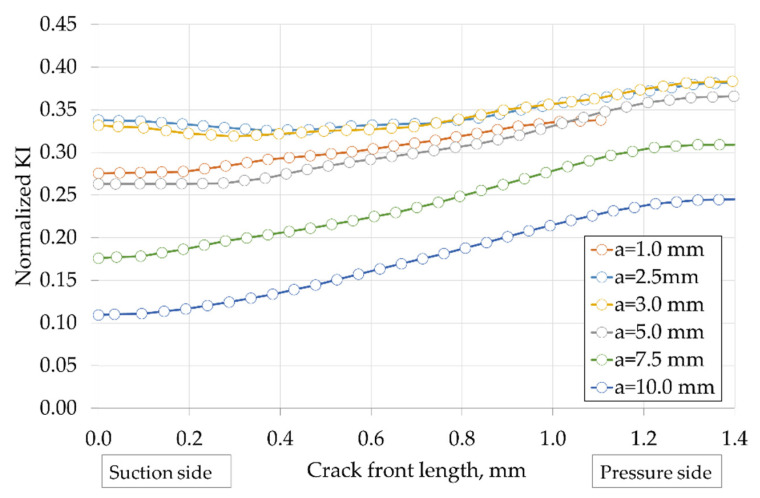
Evolution of $K_{I}$ along the crack front for a given crack length at the PETE—normalized with the material fracture toughness.

**Figure 21 materials-15-04694-f021:**
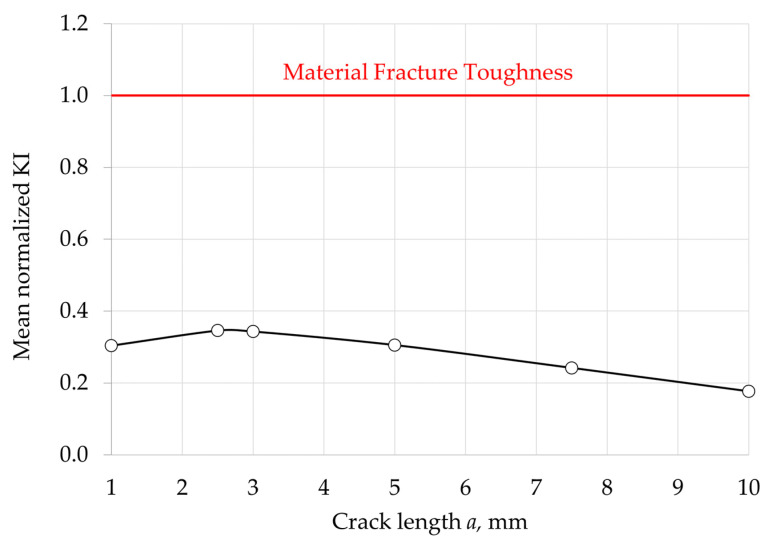
Evolution of $K_{I}$ with the crack length a at the PETE—normalized with the material fracture toughness.

**Figure 22 materials-15-04694-f022:**
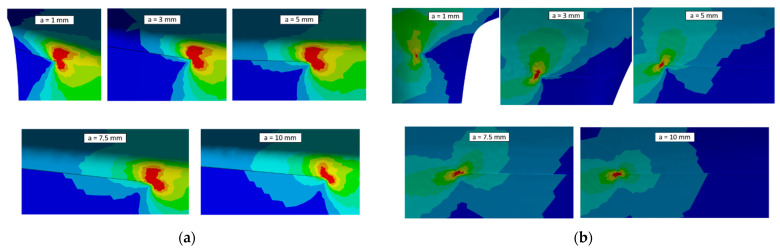
Equivalent von Mises stress analysis results. (**a**) Distribution of the stress around the PETE on the pressure side of the airfoil at the different crack sizes; (**b**) distribution of the stress around the PETE on the suction side of the airfoil at the different crack sizes.

**Figure 23 materials-15-04694-f023:**
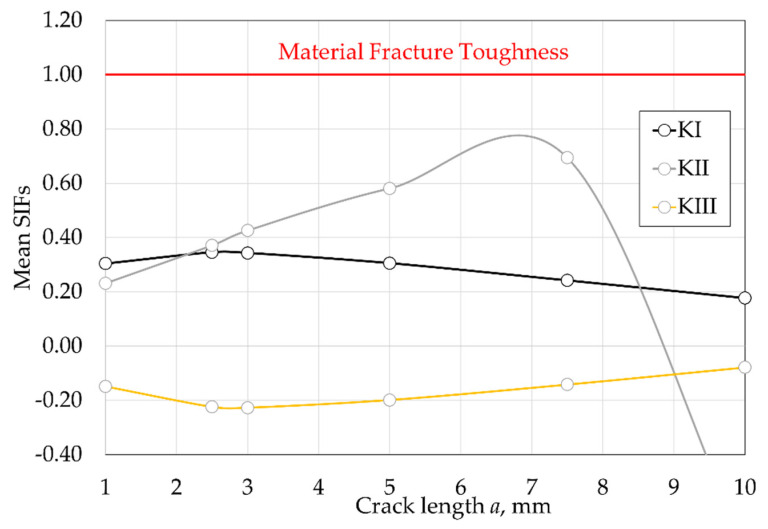
Evolution of the SIFs with the crack length a at the PETE of the vane compared and normalized with the material fracture toughness.

**Figure 24 materials-15-04694-f024:**
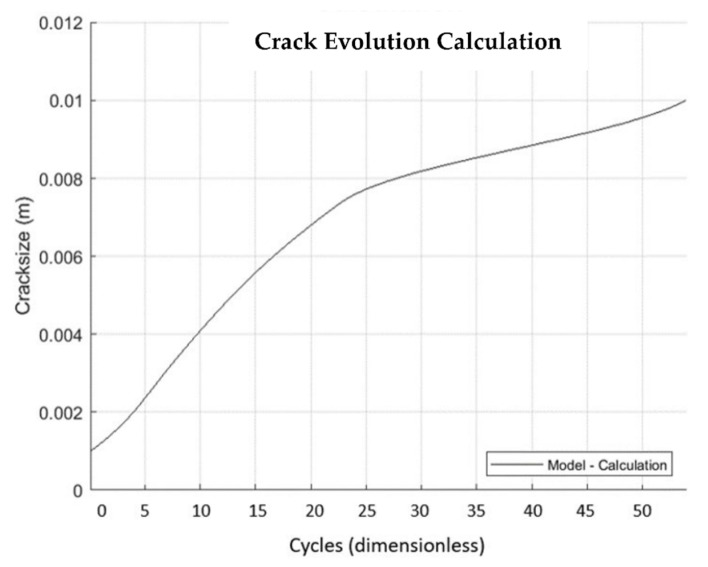
Crack growth evolution with the number of cycles at an operating cycle of ten firing hours.

**Figure 25 materials-15-04694-f025:**
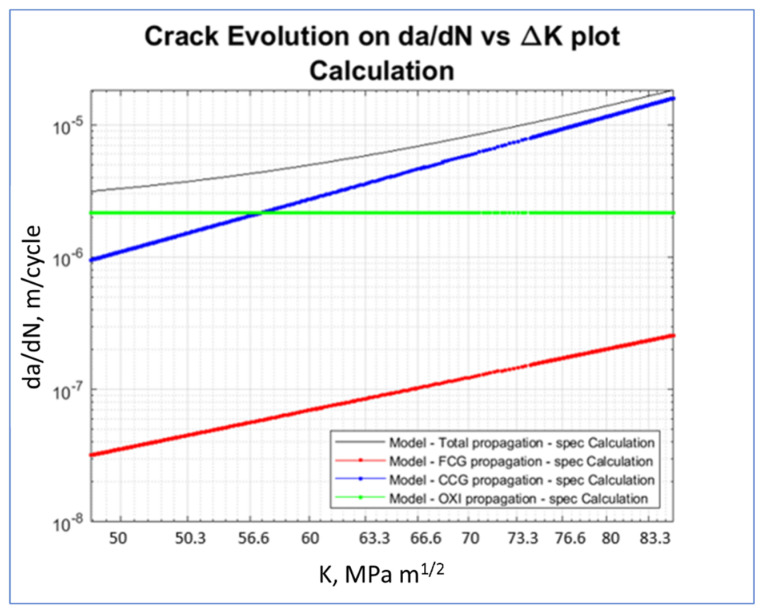
Crack growth evolution in terms of SIFs at an operating cycle of ten firing hours.

**Figure 26 materials-15-04694-f026:**
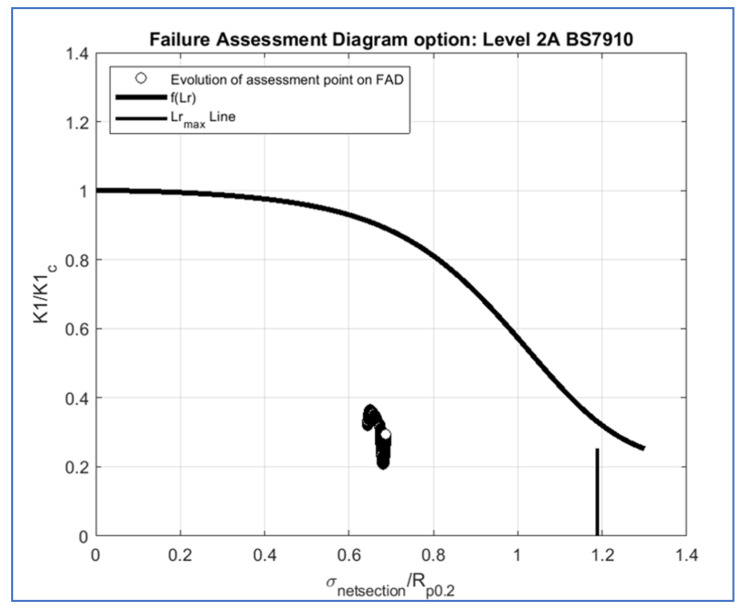
Failure assessment diagram (FAD) of the cracked component.

**Figure 27 materials-15-04694-f027:**
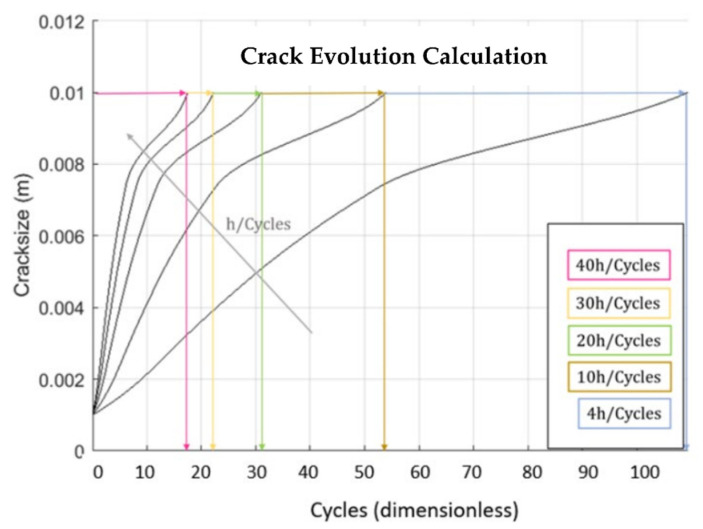
Crack Evolution at the different operating cycles.

**Figure 28 materials-15-04694-f028:**
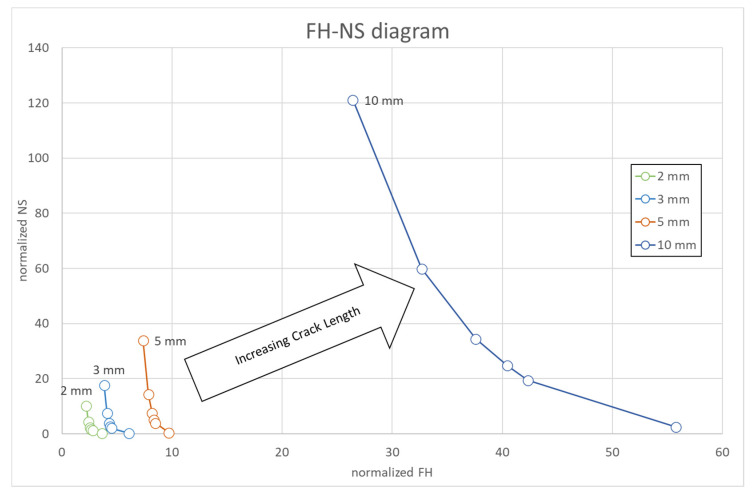
FH–NS diagram at the different crack sizes.

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
