# Peer review of "Linear Elastic Fracture Mechanics Assessment of a Gas Turbine Vane"

_materials, 2022, doi:10.3390/ma15134694_

Round 1
Reviewer 1 Report
Gas Turbines are widely used in energy, power engineering. However, its damage mechanisms of GT play an important role in crack initiation and propagation under extremely harsh operating conditions. Authors investigated the linear elastic fracture mechanics of GT vane by finite element analysis. Some results have been carried out. However, the current form of this study cannot be acceptable. Some aspects as listed below:
1. In Fig.4, more details about the temperature distribution should be given. Here, the temperature is measured or predicted?
2.Fig 13 should be improved.
3. In Fig14 and Fig15, the results unit should be given.
4. More discussion about the crack analysis should be given.
Author Response
See the annexed file

Reviewer 2 Report
The manuscript focused on the crack propagation at the most critical point of a second stage of a gas turbine blade by means of the Linear Elastic Fracture Mechanics. Advancing the knowledge on these topics is relevant for the corresponding research community. In general, the manuscript gives a good description of the proposed methodology and results and conclusions are supported by the empirical data analyzed in the study. Please provide the information about calculation number, simulation time and computational cost of the proposed method. Please explain the superiority of the proposed method through comparison with other methods.
Author Response
See the annexed file

Reviewer 3 Report
The authors have performed an evaluation of the linear elastic fracture mechanics of a specific gas turbine using finite element analysis. By means of numerical simulations they evaluate the crack propagation at the most critical point of a second stage of a gas turbine blade. They compare the numerical results with the analytical solution to evaluate the crack growth due to fatigue stress and creep. The article is relevant to the subject of fracture mechanics and suitable for the journal.
After analyzing the authors' work in detail, I recommend some modifications to the structure of the article and I would also like the authors to answer some curiosities:
1. what reference does the authors make to guarantee the service life of the in-service behavior of the component?
2. With which criteria have been established in case safety factors have been established for the thermomechanical loads used in the numerical simulation.
3. More theoretical basis of fracture mechanics is missing.
4. Expand the impact references in the introduction section.
5. The discussions do not relate the results with findings obtained by other researchers.
6. Expand the conclusions with some sentence or small paragraph commenting on where research can be directed in the line of investigation of the article and what all these advances in the in-service behavior of the gas turbine vane can mean.
Author Response
See the annexed file

Reviewer 4 Report
The article highlights peculiarities of crack assessment on a second-stage gas turbine vane with crack propagation analysis in terms of fatigue, creep and oxidation.
The article is interesting, but a number of shortcomings need to be corrected:
- It is necessary to change the numbering of subsection 1.2 (line 30) to 1.1.
- The sentences in the Introduction section (Lines 101-105) should be moved to the Numerical model section.
- The sentences in the Introduction section (Lines 105-107) should be moved to the Results section.
- In Figs. 3a, 4a, 6, 13, 14, 15, 17a, 18, 19, 21, 22, 23, 24, 25, 26, 27, and 28, the font size should be increased.
- References to Figures 7, 8, 9, 17, 19, 22, 23, 25, and 27 were given after the figures in the manuscript. However, the reference should be given in the text before corresponding figure. This should be corrected.
- It is necessary to indicate for what reasons the initial crack size (in this analysis was of 1 mm) and a cyclic target (3000 cycles) was chosen.
- In Fig. 16, the values of creep strain should be given.
- The material of a gas turbine vane for which the simulation is performed should be specified.
- In Fig. 21, the decrease in ?? in terms of fracture mechanics should be explained in detail. In the analysis of Figs. 20, 21 and 23 authors should pay attention to the conditions ensuring plane strain or plane stress. Ensuring the same conditions of plane strain or plane stress for crack lengths of 1…10 mm (??≈25…90) is problematic.
- In Figs. 20 and 21, КІ units should be provided.
- In Fig. 23, SIF units should be provided.
- In Fig. 25, K units should be given.
- In Figs. 24 and 27, the crack length should be given in mm.
Author Response
See the annexed file

Reviewer 5 Report
The abstract is OK
From the abstract and introduction there is not clear which is the novelty of this work; rather it seems an extension of previous work! Also please highlights the scientific novelty of this work. Because the fracture mechanics is well developed field
You consider here a crack that nuclear but what about having multiple site of nucleation of crack ?
There is need detail about boundary condition ! mesh details are missing too
You spoke about creep, oxidation but no details were presented- how to replicate this !
It is understandable about confidentially and also about normalization but not clear how to interpret your normalization and your results !
Because for example Figure 4 shows the temperature but not clear about what you spoke about there – as well this was used for creep ?
“The region of main interested is finely modelled” this is not part of a research paper – because it requires details in order to replicate and interpret
“Mechanical loads (pressure distribution) directly applied on the sub model.” As above – please clarify and make the necessary things in order to understand the model/case proposed for analysis
Figure 8 probably requires a citation
Eq 4 is to general and not incorporate any your case loading ( creep, oxidation and so on )
From propagation criterion seems that you only consider the elastic part without any plastic one so how can be linked to damage ?
The discussion should be slightly better elaborated against some literature data
The conclusions requires some quantitative details rather then verbatim
Author Response
See the annexed file

Round 2
Reviewer 3 Report
The authors have answered all questions with very good arguments. I have nothing more to add. The article has improved in quality.
Author Response
Many thanks
Reviewer 4 Report
The authors took into account almost all comments of the reviewer and made appropriate corrections to the manuscript.
However, a number of shortcomings need to be corrected:
1. In Figs. 1, 13 should be increased the font size.
2. In Fig. 25, K units should be given (The reviewer did not see the difference between the old and the new version of this figure).
Author Response
Font size in figures 1 and 13 have been increased;
Units in figure 25 have been added.
Reviewer 5 Report
-
Author Response
many thanks